# Cardiac-Specific Suppression of Valosin-Containing Protein Induces Progressive Heart Failure and Premature Mortality Correlating with Temporal Dysregulations in mTOR Complex 2 and Protein Phosphatase 1

**DOI:** 10.3390/ijms25126445

**Published:** 2024-06-11

**Authors:** Xiaonan Sun, Xicong Tang, Hongyu Qiu

**Affiliations:** 1Center for Molecular and Translational Medicine, Institute of Biomedical Science, Georgia State University, Atlanta, GA 30303, USA; sxn1220@gmail.com (X.S.); tang9@arizona.edu (X.T.); 2Cardiovascular Translational Research Center, Department of Internal Medicine, College of Medicine-Phoenix, University of Arizona, Phoenix, AZ 85004, USA; 3Clinical Translational Sciences (CTS) and Bio5 Institution, University of Arizona, Tucson, AZ 85721, USA

**Keywords:** valosin-containing protein, heart failure, mTOR complex, protein phosphatase 1

## Abstract

Valosin-containing protein (VCP), an ATPase-associated protein, is emerging as a crucial regulator in cardiac pathologies. However, the pivotal role of VCP in the heart under physiological conditions remains undetermined. In this study, we tested a hypothesis that sufficient VCP expression is required for cardiac development and physiological cardiac function. Thus, we generated a cardiac-specific VCP knockout (KO) mouse model and assessed the consequences of VCP suppression on the heart through physiological and molecular studies at baseline. Our results reveal that homozygous KO mice are embryonically lethal, whereas heterozygous KO mice with a reduction in VCP by ~40% in the heart are viable at birth but progressively develop heart failure and succumb to mortality at the age of 10 to 12 months. The suppression of VCP induced a selective activation of the mammalian target of rapamycin complex 1 (mTORC1) but not mTORC2 at the early age of 12 weeks. The prolonged suppression of VCP increased the expression (by ~2 folds) and nuclear translocation (by >4 folds) of protein phosphatase 1 (PP1), a key mediator of protein dephosphorylation, accompanied by a remarked reduction (~80%) in AKTSer473 phosphorylation in VCP KO mouse hearts at a later age but not the early stage. These temporal molecular alterations were highly associated with the progressive decline in cardiac function. Overall, our findings shed light on the essential role of VCP in the heart under physiological conditions, providing new insights into molecular mechanisms in the development of heart failure.

## 1. Introduction

Valosin-containing protein (VCP), also known as p97 in mammals, Cdc48 in yeast and plants, CDC-48 in worms, and Ter94 in flies, is an ATPase-associated protein ubiquitously expressed in cells [1,2,3,4]. By interacting with adaptor proteins, VCP has been associated with many cellular pathways, including transcriptional regulation, protein degradation, and cellular stress responses [1,4,5,6,7,8,9,10,11,12,13,14,15,16,17,18,19,20]. Specific genetic mutations of VCP are associated with severe dilated cardiomyopathy [21] and a multisystem degenerative disorder termed IBMPFD/ALS, encompassing muscle, brain, and bone and other diseases, such as Parkinsonism [22,23,24,25,26]. We previously identified that VCP is expressed in cardiomyocytes in vitro and in mouse hearts in vivo [27,28], and the downregulation of VCP under stress conditions is highly associated with cardiac dysfunction. The cardiac-specific overexpression of VCP protects the heart against ischemic injury and pressure overload (PO)-induced heart failure [16,27,28,29,30,31,32]. These results together indicate that VCP is an important regulator in cardiomyocytes and plays a protective role in cardiac pathological and stress conditions. However, the essential role of VCP in maintaining cardiac function under physiological conditions has yet to be fully elucidated.

It is known that the mammalian target of rapamycin (mTOR) signaling is a key pathway regulating cardiac growth and survival in both physiological and pathological conditions [33,34,35,36,37,38,39]. By binding or connecting with multiple different co-factors/adaptors, mTOR plays distinctive roles in various cells through the constituted complexes, such as mTOR complex 1 (mTORC1) and mTORC2. It has been shown that mTORC1 regulates protein synthesis, cell growth and proliferation, autophagy, cell metabolism, and stress responses, whereas mTORC2 appears to regulate cell survival and contractility [33,40,41]. Our previous studies showed that, under cardiac stress caused by PO, a reduction in VCP is accompanied by an elevated mTORC1 activity and a down-regulated mTORC2 [42], indicating a dual role of VCP in regulating mTOR complexes under pathological conditions. However, whether VCP expression is necessary to regulate mTOR signaling in the heart at physiological status is unknown.

In the present study, we tested the hypothesis that a sufficient VCP expression is required for cardiac development and maintaining cardiac function; we posited reducing VCP in cardiomyocytes would dysregulate mTOR signaling and impair cardiac function. As shown in Figure 1, to examine this hypothesis, we developed a cardiac-specific VCP knockout (KO) mouse model and compared the induced consequences in these KO mice to their litter-matched control mice through a complementary approach; first, we conducted the in vivo physiological studies to determine whether the suppression of VCP in cardiomyocytes impaired cardiac development and function. Additionally, we established a time-course study to investigate whether VCP insufficiency triggered the onset of cardiomyopathy and drove heart failure at baseline, akin to what we observed in a stressed heart; secondly, we performed the ex vivo histological studies to assess the heart alterations caused by VCP deletion at the tissue level; thirdly, we conducted in vitro studies to explore the molecular regulations of deficient VCP on mTOR signaling and potential underlying mechanisms, using the heart tissues isolated from these mice.

Using this cell-specific VCP KO mouse model, our study unveiled a previously uncharacterized phenotype in mice with suppressed VCP expression in the heart, offering novel insights into the essential role of VCP in cardiac function under physiological conditions. Furthermore, our investigation illustrated the temporal molecular regulation of VCP on mTOR complexes and the expression and nuclear translocation of protein phosphatase 1 (PP1), a crucial mediator of dephosphorylation in the mouse heart. These findings shed new light on the cardiac response to deficient VCP, contributing to a better understanding of the molecular mechanisms underlying cardiac deterioration under stress and in age-related cardiomyopathy.

## 2. Results

### 2.1. Sufficient VCP in Cardiomyocytes Is Fundamental for Mouse Development and Survival

To elucidate the essential role of VCP in cardiac growth and function, we initially aimed to develop a cardiac-specific homozygous KO mouse that completely deleted VCP from cardiomyocytes, as detailed in the Methods Section. However, none of these homozygous VCP KO mice were viable, suggesting that cardiomyocyte VCP is fundamentally required for cardiac development and mouse survival.

Subsequently, we developed a VCP heterozygous KO mouse by partially reducing VCP expression in the cardiomyocytes. As illustrated in Figure 2A, we first generated a VCP flapped mouse (VCP f/+), in which exons 4 and 5 were targeted, and then this VCP flapped mouse was bred with an αMHC Cre+/− (Cre+/−) mouse (Figure 2A), in which the Cre gene expression was under the control of an αMHC promoter, to develop a cardiac-specific VCP heterozygous KO mouse (referred to as VCP KO in this study due to no surviving VCP homozygous KO). The resulting mice were genotyped by PCR, as shown in Figure 2B. These mice were born by Mendelian distribution.

Since VCP KO mice harbor two specific genetic modifications, VCP f/+ and αMHC Cre, we also compared these KO mice to transgenic (TG) mice expressing only the signal gene to exclude any potential off-target effects of signal gene expression. In addition to the WT mouse group, we compared the VCP KO mice to those with VCP f/+ and αMHC Cre modifications, thus incorporating three control groups into our analysis.

The Western blotting analysis revealed that, compared to WT mice, there is no significant difference in VCP expression in the hearts of VCP f/+ and αMHC Cre+/− mice, while VCP KO mice exhibited approximately 40–45% reduction in VCP protein expression (Figure 2C). Consequently, all experiments in this study were conducted using these VCP KO mice and compared to their control counterparts, including VCP f/+, Cre+/−, and WT mice.

We employed echocardiography to evaluate the cardiac physiological and morphological characteristics of VCP KO mice at 4 weeks of age and compared them to their litter-matched control mice. The parameters assessed included left ventricle (LV) end-diastolic and end-systolic anterior and posterior wall thickness (LVAWd, LVAWs, LVPWd, and LVPWs), LV internal dimensions (LVEDd and LVEDs), left ventricular ejection fraction (LVEF), and fractional shortening (LVFS). As depicted in Figure 2D–F, there were no significant differences between VCP KO mice and their control counterparts at the age of 4 weeks in terms of cardiac wall thickness (LVAWd, LVAWs, LVPWd, and LVPWs) (Figure 2D), cardiac internal dimensions (LVEDd and LVEDs) (Figure 2E), and cardiac contractility, as indicated by LVEF and LVFS (Figure 2F).

However, despite the similarity in cardiac morphology and function at this early age between VCP KO mice and their control counterparts, VCP KO mice exhibited significantly higher mortality starting at 9 months of age, and ultimately all VCP KO mice died before reaching 12 months of age (Figure 2G). These data suggest that the partial suppression of VCP in the heart results in chronic deterioration in these KO mice, leading to premature mortality.

Together, our results indicate that sufficient VCP in cardiomyocytes is necessary for mouse development and survival.

### 2.2. Suppression of Cardiomyocyte VCP Results in Progressive Cardiomyopathy and Declining Contractile Function

To investigate the reasons underlying early mouse death in VCP KO mice, we conducted a time-course study to monitor cardiac physiological changes in these mice before their demise using echocardiography. We compared them to their litter-matched control mice at the ages of 12 weeks and 32 weeks.

As depicted in Figure 3A–C, the differences in cardiac morphology and contractile function between VCP KO mice and their control counterparts were less evident at 12 weeks of age. However, by 32 weeks of age, VCP KO mice exhibited notably reduced end-systolic anterior wall thickness (LVAWs) (Figure 3A) and increased cardiac internal dimension diameter (LVIDs) (Figure 3B) compared to their 12-weeks-of-age counterparts, while no significant differences were observed among the three control mouse groups. Contractile function, as indicated by LVEF and LVFS, showed a remarkable decline in VCP KO mice at 32 weeks vs. 12 weeks and dramatically lower than their control counterparts (Figure 3C), suggesting a progression of cardiac dysfunction in VCP KO mice between 12 weeks and 32 weeks of age.

The histological analyses also revealed significant differences between VCP KO mice and their control counterparts. While no significant differences were observed between VCP KO mice and control mice at 12 weeks of age (Figure 3D,E), the ratio of left ventricle (LV) to body weight was significantly higher in VCP KO mice compared to the control mice at 32 weeks of age (Figure 3D). Additionally, the ratio of lung to body weight, an index of heart failure, was also increased in VCP KO mice at 32 weeks of age (Figure 3E). These data further support the cardiomyopathy and heart failure developed in the VCP KO mice at a later age before demise.

### 2.3. Reducing VCP Selectively Activates mTORC1 but Not mTORC2 at an Early Stage with Compensatory Cardiac Function

We then delved into the molecular alterations induced by the suppression of VCP in the hearts. Given the absence of significant differences in physiological and histological characteristics among the three control mouse groups (i.e., VCP f/+, αMHC Cre+/−, and WT mice), we utilized WT as a representative control group for the molecular studies.

We first investigated the molecular regulations in VCP KO mouse hearts at 12 weeks of age, a period when cardiac function remains unchanged. We focused on the mTOR pathway due to its pivotal role in regulating cardiac growth and survival under physiological conditions.

As depicted in Figure 4A, upon the suppression of VCP in the heart, the phosphorylation of mTOR (p-mTOR) was significantly increased in VCP KO mice compared to WT mice, despite no significant changes in total mTOR expression.

Additionally, mTORC1 was activated in VCP KO mice vs. WT mice, as evidenced by an increase in the expression of raptor, a key adaptor for mTORC1 (Figure 4B), and the phosphorylation of AKT at Thr308 (pAKT308), a key downstream target of mTORC1 (Figure 4B). However, there were no significant differences between VCP KO and WT mice in the expression of Rictor, a key adaptor of mTORC2 (Figure 4C), or the phosphorylation of AKT at Ser473 (pAKT473), a key downstream target of mTORC2 (Figure 4C), indicating no significant impact in mTORC2 upon VCP suppression at 12 weeks of age.

These findings collectively suggest that VCP suppression in the heart selectively activates mTORC1 but not mTORC2 at an early stage when cardiac function remains unchanged, hinting at a potential compensatory mechanism.

### 2.4. Prolonged Suppression of VCP Inhibits p-AKT473 and Increases PP1 and p47

We subsequently investigated the molecular basis of the cardiac dysfunction in VCP KO mice at 32 weeks of age, when heart function failed.

As depicted in Figure 5A, upon the suppression of VCP expression (Figure 5B), VCP KO mice exhibited a pronounced reduction in p-AKTs473 versus WT mice (Figure 5C), while the total AKT expression was comparable between the two groups (Figure 5D), leading to a significant decrease in the p-AKT473/AKT ratio (Figure 5E). These results indicate a substantial inhibition of AKT Ser473 phosphorylation in the hearts of VCP KO mice at 32 weeks of age.

Next, we explored the potential mechanism that caused the reduction in pAKT473 in the VCP KO mouse hearts. As depicted in Figure 5F, upon the suppression of cardiomyocyte VCP, there is a significant increase in PP1, a key enzyme mediating protein dephosphorylation, in VCP KO mouse hearts compared to WT mice (Figure 5G). Additionally, p47, a co-factor of VCP driving inhibitory effects on ATPase, was significantly elevated in VCP KO mouse hearts compared to WT (Figure 5H). These results indicate a potential link between increased PP1 and p47 and reduced pAKT473.

### 2.5. Deficient VCP Expression in Cardiomyocytes Results in Temporal Alterations in the Subcellular Distribution of VCP and PP1

We further investigated the potential mechanisms underlying the discrepancy of pAKT473 observed in VCP KO mice between 12 weeks and 32 weeks by examining the subcellular distribution of the key proteins in the heart tissues of VCP KO mice.

As shown in Figure 6A,B, at 12 weeks of age, the partial deletion of VCP in cardiomyocytes led to a lower VCP level in both nuclear and cytoplasmic fractions of mouse heart tissues in VCP KO mice compared to WT mice. Still, there was a more pronounced reduction in the nuclear fraction (Figure 6C). Additionally, we observed that PP1 was predominantly located in cytoplasmic fractions, with no significant difference between the two groups at 12 weeks of age (Figure 6D).

However, as presented in Figure 6E,F, by 32 weeks of age, VCP expression was mainly reduced in cytoplasmic fractions, with a slight increase in the nuclear fraction of mouse hearts from VCP KO versus WT mice (Figure 6G), indicating an enhanced VCP nuclear translocation in VCP KO mouse hearts at a later age.

Interestingly, compared to WT mice, VCP KO mice exhibited a remarkably increased PP1 distribution in the nuclear fraction of mouse heart tissues at 32 weeks of age (Figure 6H). These findings suggest that the prolonged suppression of VCP in the heart not only increases PP1 expression but also enhances its nuclear translocation at a later age.

Our results indicate that VCP KO mice consistently display an increased nuclear translocation of VCP and PP1 in hearts at later ages, coinciding with AKT’s dephosphorylation at Ser473.

## 3. Discussion

This groundbreaking study represents the first utilization of a cardiac-specific VCP KO mouse model to elucidate the impact of VCP deficiency on the heart in a physiological context. The findings from this investigation underscore the essential role of VCP in mouse development and the maintenance of physiological cardiac function under unstressed conditions. Moreover, our results confirm a previously unidentified role of VCP as a negative regulator of mTORC1 in the heart and unveil a distinct effect of VCP on mTORC1 and mTORC2 during cardiac compensatory stages. Additionally, our study uncovers a previously unrecognized impact of cardiomyocyte VCP deficiency on the regulation of PP1 expression and nuclear translocation in the heart, highlighting a potential link between the increased PP1 and decreased p-AKT473 in the heart. Taken together, our results unveil a potential novel mechanism underlying heart failure induced by VCP deficiency, which is closely associated with the dephosphorylation of p-AKT473 and the activation of PP1. Overall, this investigation offers novel insights into the essential role of VCP in the heart, expands our understanding of the regulatory role of VCP in cellular signaling pathways, and sheds light on the potential mechanisms underlying heart failure.

It has been documented that genetic mutations or the functional inhibition of VCP induce chronic cardiomyopathy and cardiac dysfunction [21,43]. Our previous studies have also indicated a strong correlation between decreased VCP expression and PO-induced heart failure [30,31,32,42]. These findings underscore the association of cardiac VCP insufficiency with cardiomyopathy and stress-induced cardiac dysfunction. However, the impact of deficient VCP expression on cardiomyocytes under physiological conditions remains less understood. In contrast to most previous experimental animal models focusing on specific VCP gene mutations to investigate genetic defect-induced pathological conditions, our current study is the first to concentrate on the consequences of insufficient VCP expression in heart development and function maintenance at physiological conditions. Our results reveal that the complete deletion of VCP in cardiomyocytes leads to embryonic lethality, indicating the fundamental requirement of VCP for cardiac development and mouse survival. While the heterozygous VCP KO mice with a partial reduction in VCP in cardiomyocytes are viable at birth, they progressively exhibit impaired heart contractile activity, eventually culminating in heart failure and premature death at a young age. These findings collectively suggest that an adequate VCP expression is indispensable for cardiac growth and survival under physiological conditions.

It is widely recognized that mTOR signaling plays a pivotal role in regulating cardiac growth and survival under both physiological and pathological conditions [33], yet the mechanisms regulating this signaling in the heart remain to be discovered. Our previous studies have demonstrated that, during the early stage of PO stress, decreased VCP level coincides with the activation of mTORC1 signaling in the stressed heart. However, it remains unclear whether these alterations in mTORC1 result from PO stress or are caused by VCP reduction. Our current study addressed this question by investigating the impact of VCP suppression under a condition without stress. In conjunction with our previous data, our results confirm that VCP acts as an inhibitory regulator of mTORC1 under both stressed and unstressed conditions. It also suggests that the activation of mTORC1 may represent an initial compensatory response to VCP decrease in the heart aimed at preserving cardiac function. Notably, despite the increase in phosphorylated mTOR (p-mTOR), no significant changes were observed in mTORC2 signaling at 12 weeks, indicating a selective regulation of VCP on mTORC1 and C2 at this period.

Interestingly, our results further reveal that the prolonged suppression of VCP significantly decreased phosphorylated AKT at Ser473 (p-AKT473), a key downstream signal of mTORC2, at 32 weeks of age. These findings suggest a temporal effect of VCP on mTORC2 in unstressed hearts. Notably, the temporal alterations in p-AKT473 were accompanied by consistent phenotypic changes in cardiac function in VCP KO mice. These data highlight a strong association between p-AKT473 and cardiac function in VCP KO mice. Our previous study also supports the essential role of p-AKT473 in the cardioprotective effects conferred by VCP [27]. For example, our previous in vitro studies demonstrated that the overexpression of VCP increased p-AKT473 in cardiomyocytes, thereby protecting cardiomyocytes from stress-induced death. Moreover, this VCP-mediated protection was abolished by an Akt inhibitor [27,28,29]. These in vitro findings underscore the vital role of VCP/p-AKT473 in determining cardiomyocyte survival. Taken together, our results highlight a mechanism of heart failure in VCP KO mice that may be caused by the repression of VCP/p-AK473 signaling in cardiomyocytes.

In addition, our results reveal that PP1 exhibits a similar temporal change in response to VCP suppression in VCP KO mouse hearts. PP1, the predominant dephosphorylating enzyme accounting for up to 70% of the entire serine/threonine phosphatase activity in the heart [44,45,46], plays a crucial role in the regulation of cardiac excitation-contraction coupling [47,48]. Accumulating evidence indicates that PP1 expression and activity are increased both in the human failing heart and in experimental models of heart failure [49,50,51,52,53,54,55]. Notably, a threefold heart-specific overexpression of PP1a was sufficient to induce severe cardiomyopathy in mice [3,56]. Importantly, PP1 has been shown to interact with and directly dephosphorylate AKT at S473 in an in vitro phosphatase assay using purified PP1 and AKT proteins [57,58]. Thus, the increased PP1 may contribute to decreased p-AKT473 in the heart of VCP KO mice. Furthermore, our findings indicate that PP1 significantly increases in the nuclear fraction, where mTORC2 is abundant. We also observed a slight increase in the nuclear distribution of VCP at 32 weeks despite the decrease in total VCP expression. These coincidental increases in the nuclear translocation of VCP and PP1 may enhance the assembly of the VCP and other co-factor proteins to constitute the VCP-PP1-mTORC2 complex, thereby increasing the activation of PP1, leading to the dephosphorylation of AKT S473.

Furthermore, our results show increased p47 in VCP KO mouse hearts at the end stage. Studies have demonstrated that VCP cooperates with the SEP-domain adapters (p37 or UBXN2A) and p47 in stripping inhibitor-3 (I3) from PP1 for activation [46,59]. The increase in p47 is critical, as p47 is the most abundant of VCP adapters (with about 100-fold over p37 or UBXN2A) and also has the structural features to direct p47 to the PP1 complex for disassembly by VCP [60]. Therefore, increased p47 could facilitate the constitution of the VCP-p47-PP1 complex, activate PP1, and enhance its dephosphorylating function on p-AKT473.

Previous studies have also indicated that VCP favors binding to p47 under ATP-limiting conditions [61]. Consequently, binding to p47 further suppresses the ATPase activity of VCP [43,62], leading to a vicious circle in VCP-p47 binding and the inhibition of VCP ATP activity. These results suggest a potential mechanism of progressive cardiac deterioration in VCP KO mice. It is plausible that a decrease in VCP expression in the KO mouse reduces the ATPase activity of VCP at a young age, leading to VCP bound to p47, which in turn inhibits ATPase activity, resulting in cardiac dysfunction in older VCP KO mice. These studies suggest that increased p47 introduces an additional facet to the regulatory mechanism underlying cardiac dysfunction in VCP KO mice. Together, the main findings in the molecular regulation of VCPKO mice are summarized in Figure 7.

Despite these new findings, there are several limitations to and future directions for this study: (1). While this study demonstrated alterations in the subcellular distribution of VCP and PP1 in VCP KO mice, functional assays correlating these distributions with specific cellular outcomes would enhance our understanding of VCP-related pathologies. Assessing mitochondrial function, energy metabolism, autophagic flux, and apoptosis could provide mechanistic insights into how VCP deficiency impacts cardiomyocyte health. (2). The study convincingly shows that VCP deficiency leads to the selective activation of mTORC1. However, further investigations are needed to delve into the mechanisms underlying this selective activation. Exploring whether alterations in VCP co-factor expression and binding dynamics contribute to this phenomenon, along with investigating the involvement of upstream regulators of mTORC1 in the context of VCP deficiency, could enhance our understanding of this issue. (3). The increase in p47 in VCP KO hearts is highlighted as a critical factor in the activation of PP1. While novel and interesting, functional studies demonstrating the direct interaction and impact of p47 on PP1 activity and AKT dephosphorylation in cardiomyocytes would strengthen this conclusion. Additional experiments, such as the overexpression or knockdown of p47 in cardiomyocytes, would provide valuable insights into its role in this setting. (4). Considering the translational aspect of our findings is crucial for identifying therapeutic interventions that could benefit patients. Exploring therapeutic avenues such as mTORC1 inhibitors or AKT activators in VCP-deficient models holds promise for modulating aberrant signaling pathways implicated in VCP deficiency and may offer new opportunities for intervention.

In summary, the current study presents a novel animal model for investigating the biological function of VCP in the heart under physiological conditions and associated heart diseases. Our findings demonstrate that VCP in cardiomyocytes serves as a necessary regulator essential for heart development and the maintenance of cardiac function. Moreover, our results offer novel insights into the regulation of VCP on mTORC2 and its correlation with PP1 in response to VCP suppression, thereby deepening our understanding of the comprehensive regulation of VCP on mTOR signaling in both physiological and pathological cardiac conditions. With this unique mouse model, our study provides novel evidence elucidating potential mechanisms associated with the development of cardiomyopathy and cardiac dysfunction induced by VCP deficiency. These findings contribute to the growing body of knowledge on VCP’s role in cardiac biology and may pave the way for developing targeted therapeutic strategies for heart diseases associated with VCP dysfunction.

## 4. Materials and Methods

### 4.1. Generation of Cardiac-Specific VCP KO Mice

To develop cardiac-specific VCP KO mice, we first generated a conditional knockout mouse model (VCP flapped mouse) using a Cas9/sgRNA plasmid construction system and UCA^TM^ (Universal CRISPR Activity Assay). After scanning the mouse VCP gene structure and the size of exons, we designed a construct by selectively targeting exons 4 and 5, and the loxP elements were inserted at the intron between 3–4 and 5–6 (as shown in Figure 2A). The validated plasmid was injected in zygotes of C57BL/6N wild-type (WT) mice. The founders were identified and confirmed by PCR as VCP f/+ mice.

Then, this VCP flapped mice (VCP f/+) were cross-mated with αMHC Cre+/− mice (C57BL/6N) mice (purchased from Jackson Lab, Sacramento, CA, USA) to generate heterozygous VCP KO (VCP f/+/αMHC Cre+/−) mice (Figure 2A). The offspring were identified by PCR analysis (Figure 2B). In addition, we also attempted to generate the cardiac-specific VCP homozygous KO mice by cross-mating the VCP f/+ mice with VCP f/+/αMHC Cre+/− mice. While the heterozygous VCP KO mice survived, VCP homozygous KO mice suffered embryonic death.

Thus, all the experiments were performed on the heterozygous VCP KO mice in this study (referred to as VCP KO) and compared to the litter-matched control mice, including VCP f/+ (or VCP flapped), αMHC Cre+/−, and WT mice. Both male and female mice, aged between 4 and 52 weeks, were included in this study.

All animal procedures were performed in accordance with the NIH guidance (Guide for the Care and Use of Laboratory Animals, revised in 2011), and the protocols were approved by the Institutional Animal Care and Use Committee of Georgia State University.

### 4.2. Echocardiography

All animals were anesthetized by using 2% isoflurane. Cardiac function and structure were determined by echocardiography using a Vevo 3100 high- resolution micro-ultrasound system (FUJIFILM Visual Sonics Inc., Toronto, ON, Canada) with a 13-MHz probe. The M-mode recording was made at the middle level of the left ventricle (LV) in a left parasternal long-axis view. Cardiac structure and function indexes were measured as described previously [42,63], including left ventricle (LV) end-diastolic and end-systolic anterior and posterior wall thickness (LVAWd, LVAWs, LVPWd, and LVPWs), LV internal dimensions (LVEDd and LVEDs), LV ejection fraction (LVEF), and fractional shortening (LVFS) [42,43,64]. All the mice will be randomly assigned to the examination and blinded to investigators performing echography, with heart rates maintained between 450 and 550 mph during the echography.

### 4.3. Histology

Mouse heart tissues were collected for ex vivo assessments. The mouse hearts, LVs, and lungs were weighted and normalized by body weight [30,42,64].

### 4.4. Protein Extraction and Subcellular Fraction

Total protein and subcellular fractions were extracted as described previously [42,43]. In brief, for total protein extraction, heart tissues were homogenized with the homogenizer (OMNI, bead mill homogenizer) in RIPA buffer with EDTA (Boston bioproducts, BP-115D) and protease inhibitor (Roche,11836153001) and phosphatase inhibitor (Roche, 04906837001), then centrifuged at 14,000 rpm at 4 °C for 15 min. For subcellular fractions, the heart tissues were separated into cytoplasmic protein and nuclear protein according to the manufacturer’s protocol (Abcam, Nuclear Extraction kit, ab113474), as conducted previously [43].

### 4.5. Western Blotting

The protein expression and the phosphorylation were detected by Western blotting as conducted previously [43]. The specific primary antibodies (Ab) used in this study included anti-VCP Ab (Invitrogen, MA3-004), anti-P47 Ab (Santa Cruz, sc-365215), anti-PP1 AB (Santa Cruz, sc-7482), anti-Pan-AKT Ab (cell signaling, 4691s), anti-Phospho-Akt Ser473 Ab (cell signaling, 9271), anti-Phospho-Akt Thr308 Ab (cell signaling, 2965), anti-Phospho-mTOR Ab (Ser2448) (cell signaling, 2971), anti-mTOR Ab (cell signaling, 2972), anti-GAPDH Ab (cell signaling, 2118), anti-Histon 3 Ab (cell signaling, 4499), anti-Raptor (Santa Cruz, sc-81537), and anti-Rictor (cell signaling, 2140) and the corresponding secondary antibodies (Li-Cor, IRDye800CW, and IRDye680LT). The detection was performed by chemiluminescence (Li-cor, odyssey CLx), and bands were visualized using the Odyssey DLx Imaging System (LI-COR Biosciences, Lincoln, NE, USA) and quantified using Image Studio Ver 5.2., as previously described [43].

### 4.6. Statistical Analysis

Data are presented as means + standard error (SEM) for the number per group indicated in each figure legend. Differences among groups were determined by a two-way ANOVA followed by Tukey’s post hoc test. Comparisons between the two groups were performed using an unpaired Student’s *t*-test. A value of *p* < 0.05 was considered significant.

## Figures and Tables

**Figure 1 ijms-25-06445-f001:**
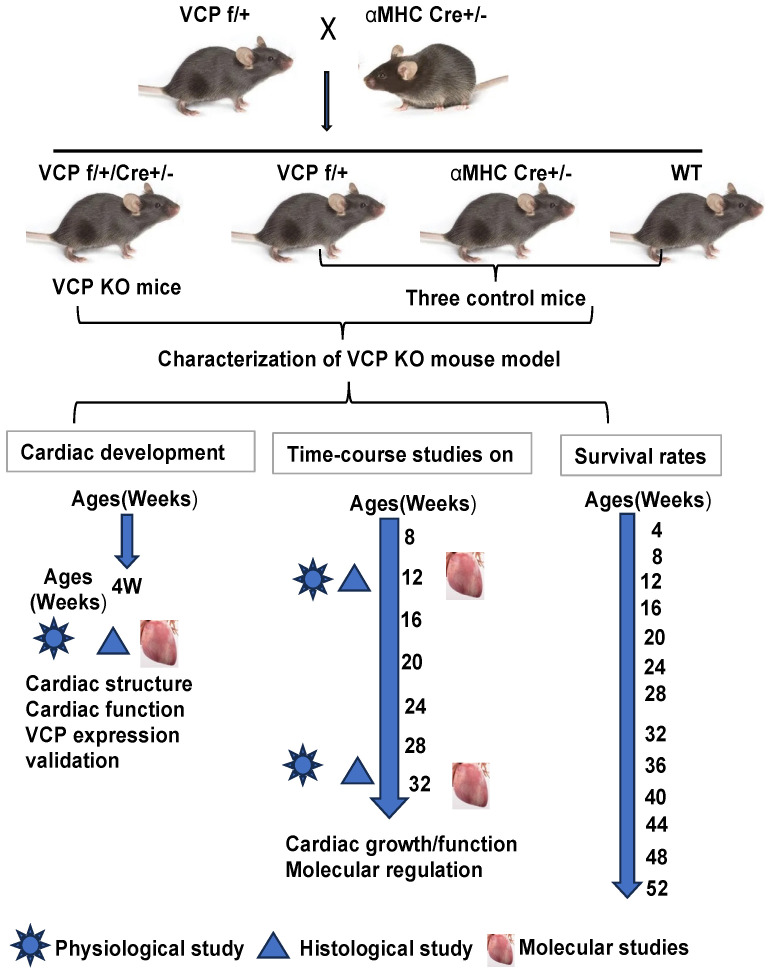
The experimental design and timelines of this study. To characterize this new model, the generated VCP KO mouse model and their three littler-matched control counterparts (VCP f/+, alpha-myosin heavy-chain (αMHC) Cre+/−, and WT) underwent serial studies to investigate the impact of VCP suppression on cardiac development and physiological function as well as mouse survival at baseline conditions.

**Figure 2 ijms-25-06445-f002:**
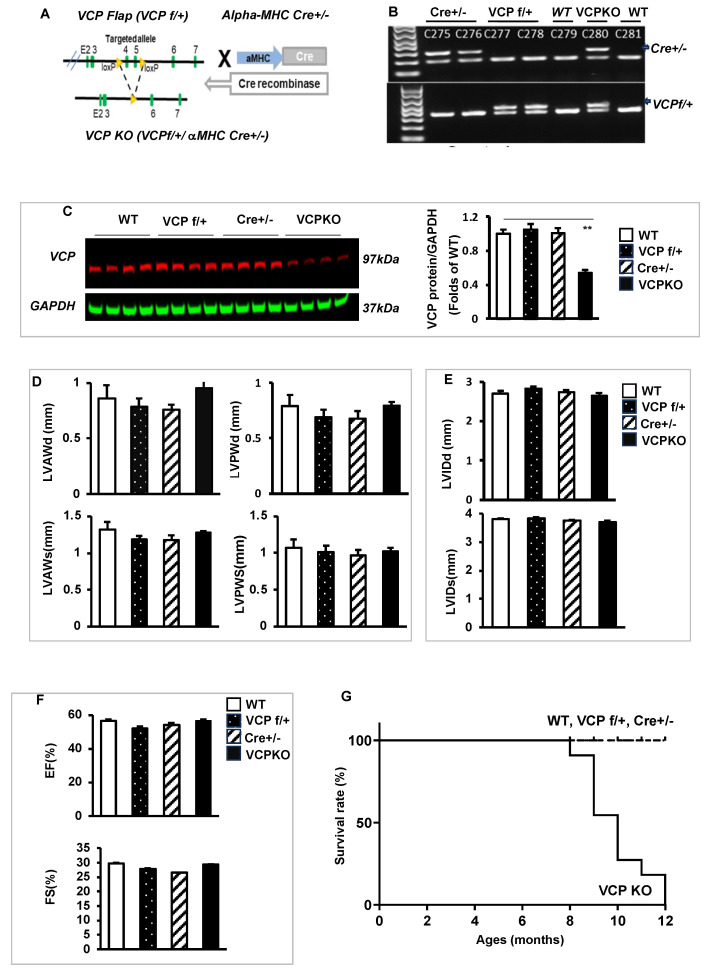
Characteristics of cardiac-specific VCP knockout (KO) mouse. (**A**). Construction and generation of VCP heterozygous KO mouse (VCP KO) by crossing VCP flapped mice (VCP f/+) and alpha MHC-Cre TG mice (αMHC Cre+/−). (**B**,**C**). Identification of mouse genotyping by PCR (**B**) and VCP protein expression in the hearts at 4 weeks by Western blots (**C**). n = 4/group; **, *p* < 0.01 vs. age-matched control mice (WT, Cre+/− VCP f/+). GAPDH was used as a loading control of proteins. (**D**–**F**). Structural and functional characteristics of VCP KO mice vs. three control mouse groups at 4 weeks (n = 6/group), including LV wall thickness (**D**), internal diameters (**E**), and contractile activity (**F**). LVAWd and LVAWs: LV anterior wall thickness at end-diastole and end-systole, respectively; LVPWd and LVPWs: LV posterior wall thickness at end-diastole and end-systole, respectively; LVEDd and LVEDs: LV internal dimensions at end-diastole and end-systole, respectively; EF and FS: LV ejection fraction and fractional shortening, respectively. (**G**). Survival curve followed up to 12 months in four groups (n = 10–11/group). The solid line represents KO mouse group, and the dotted lines represent the control mouse groups.

**Figure 3 ijms-25-06445-f003:**
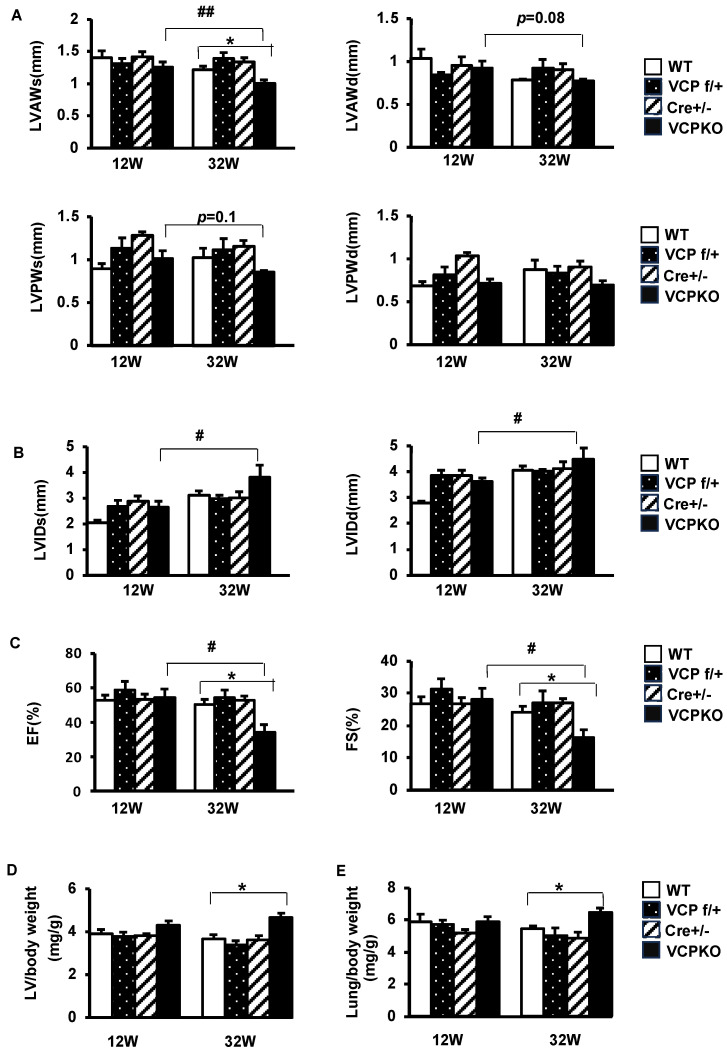
Cardiac-specific knockdown of VCP results in a progressive decline in contractile function and cardiac dilation. (**A**–**C**). Cardiac morphology and function were measured by echocardiography in four groups of mice (WT, VCP f/+, Cre +/−, and VCPKO mice) at the ages of 12 and 32 weeks, representing LV wall thickness (**A**), LV internal diameter (**B**), and LV contractile function (**C**). (**D**,**E**). Quantitation of LV weight and lung weight normalized to body weight in four groups of mice at the ages of 12 and 32 weeks. n = 6/group, *: *p* < 0.05 vs. age-matched WT, ^#^: *p* < 0.05, ^##^: *p* < 0.01 vs. the VCP KO mice at an age of 12 weeks.

**Figure 4 ijms-25-06445-f004:**
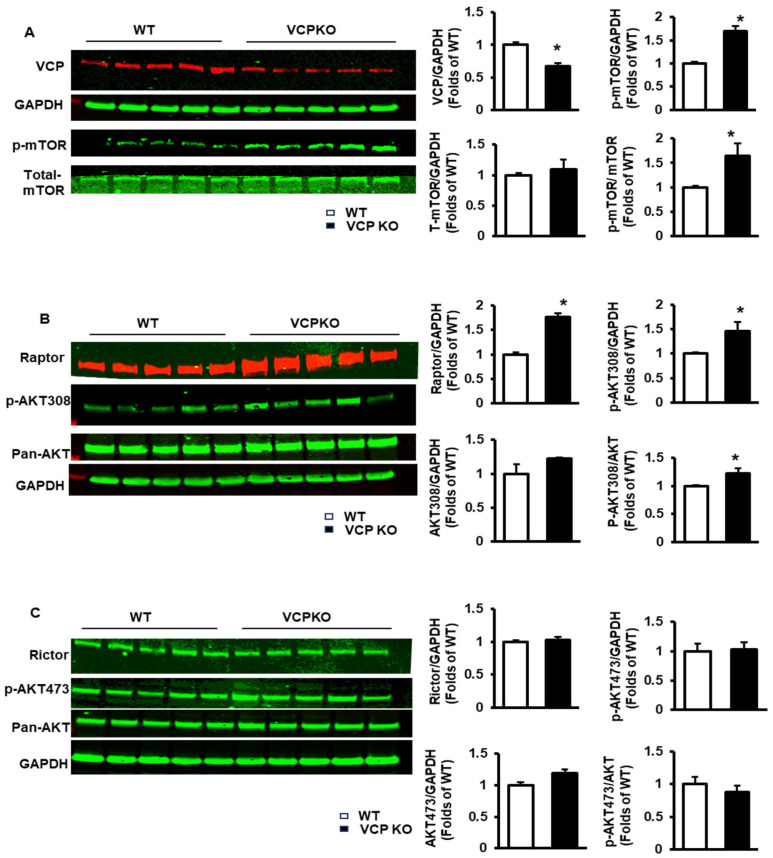
Knockdown of VCP in cardiomyocytes selectively activates mTORC1 but not mTORC2 in the heart at the age of 12 weeks. (**A**). The relative decreased VCP expression and increased mTOR phosphorylation in VCP KO mouse hearts vs. WT by Western blotting. (**B**). The activation of mTORC1 in VCP KO mice vs. WT, represented by the increased expression of mTORC1 adaptor (Raptor) and p-AKT at Thr 308 (p-AKT308), a key downstream target of mTORC1. (**C**). The expression of key mTOR C2 component (Rictor) and p-AKT at Ser473 (p-AKT473), a key downstream target of mTORC2 in the heart tissues from VCP KO mice vs. WT. n = 5/group. *, *p* < 0.05 vs. WT. GAPDH is used as a protein loading control.

**Figure 5 ijms-25-06445-f005:**
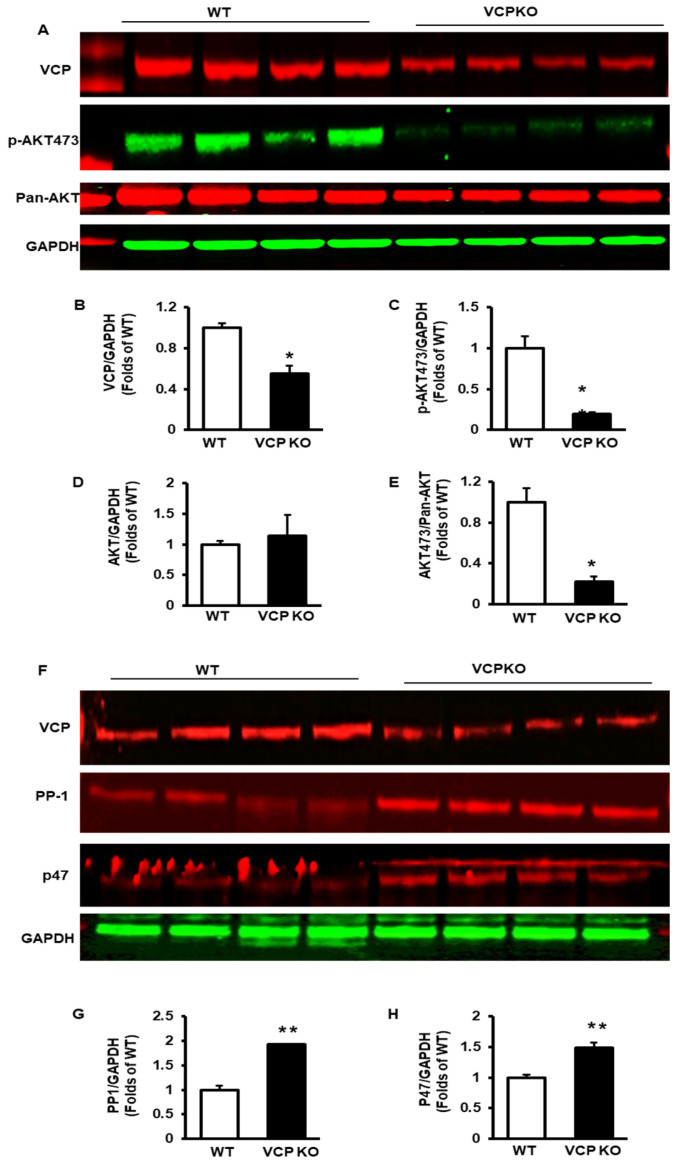
Prolonged suppression of VCP in cardiomyocytes decreases p-AKT at Ser473 and increases the expression of protein phosphatase-1 (PP1) and p47 in the hearts of VCP KO mice at the age of 32 weeks. (**A**–**E**). The representative images of Western blots (**A**) and the relative total protein value of VCP (B), p-AKT473 (**C**), and AKT (**D**), as well as the p-AKT473/AKT ratio (**E**). (**F**–**H**). The representative images of Western blots (**F**) and quantitated values of the total protein of PP1 (**G**) and p47 (**H**) in the heart tissues of VCP KO mice vs. WT. GAPDH was used as a loading control of the total protein. n = 4/group; *, *p* < 0.05 and **, *p* < 0.01 vs. age-matched WT.

**Figure 6 ijms-25-06445-f006:**
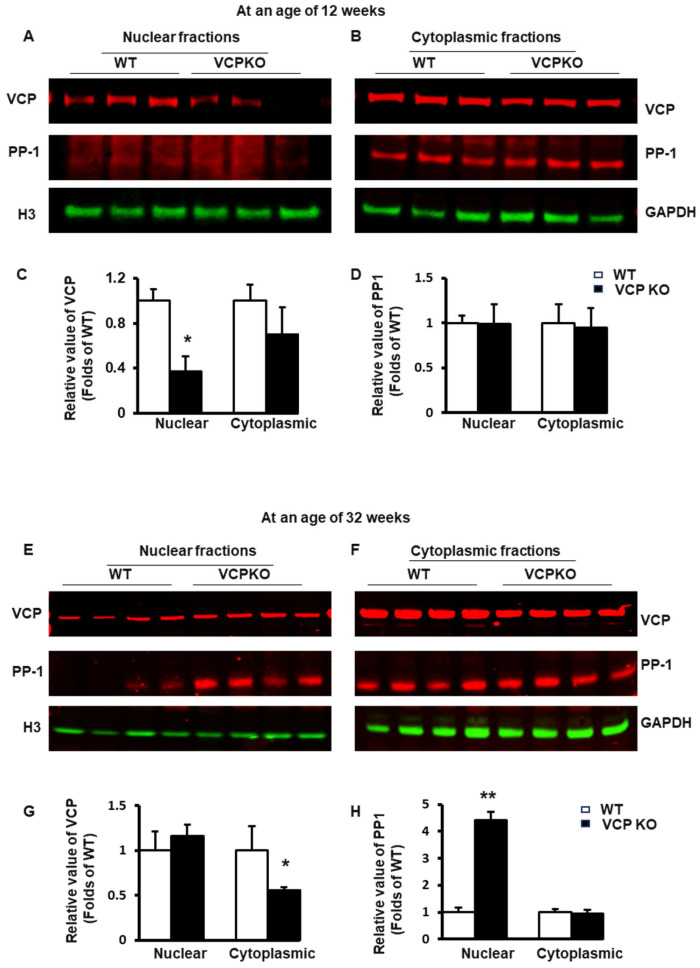
Deficient VCP expression in cardiomyocytes results in a temporal alteration in subcellular distributions of VCP and PP1 between the ages of 12 and 32 weeks in VCPKO mice. (**A**,**B**). The representative Western blots of nuclear and cytoplasmic fractions of heart tissues from VCP KO and WT mice at the age of 12 weeks. (**C**,**D**). The relative values of VCP (**C**) and PP1 (**D**) in nuclear and cytoplasmic fraction at 12 weeks. n = 3/group. (**E**,**F**). The representative Western blots of nuclear and cytoplasmic fractions of heart tissues from VCP KO and WT mice at the age of 32 weeks. (**G**,**H**). The relative values of VCP (**G**) and PP1 (**H**) in the nuclear and cytoplasmic fractions at the age of 32 weeks. n = 4/group. H3 was used as nuclear fraction loading control. GAPDH was used as cytoplasmic fraction loading control. The value of target proteins in the nuclear fractions was normalized to H3, and the value of the target proteins in cytoplasmic was normalized to GAPDH. *, *p* ≤ 0.05; **, *p* ≤ 0.01 vs. age-matched WT mice.

**Figure 7 ijms-25-06445-f007:**
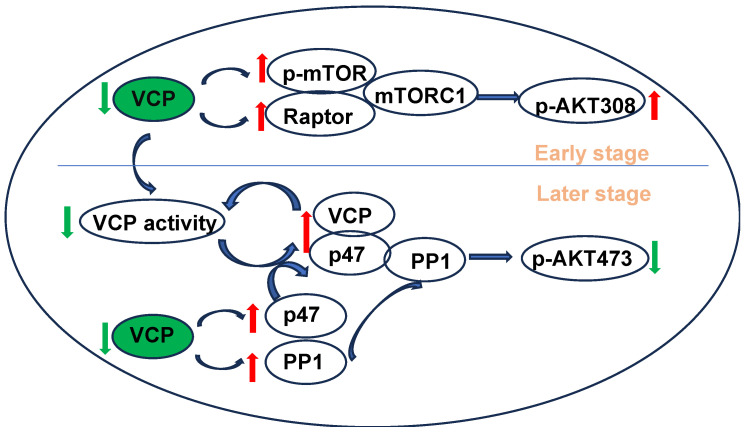
The main finding in molecular signaling caused by VCP deficiency from this study. At an early stage, the partial suppression of VCP in cardiomyocytes activates mTORC1, leading to an increase in p-AKT308. A lower VCP level decreases its ATPase activity, promoting VCP selectively binding to p47, which further inhibits ATPase activity. Moreover, the prolonged inhibition of VCP results in an increased expression of p47 and PP1, facilitating the formation of the VCP-p47-PP1 complex. This complex activates PP1 and subsequently dephosphorylating pAKTs473 at a late stage, contributing to cardiac dysfunction in older age. Green arrow: decreased, red arrow: increased, blue arrow: the resulting consequence. Green oval: VCP KO, Orange font: different stage of VCP KO.

## Data Availability

All data needed to evaluate the conclusions are presented in the paper, and all raw data and statistical p-values will be provided as requested.

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
