# Peer review of "Cardiac-Specific Suppression of Valosin-Containing Protein Induces Progressive Heart Failure and Premature Mortality Correlating with Temporal Dysregulations in mTOR Complex 2 and Protein Phosphatase 1"

_ijms, 2024, doi:10.3390/ijms25126445_

Round 1
Reviewer 1 Report
Comments and Suggestions for Authors
The authors findings emphasize essential role of VCP in the heart and its potential role in molecular mechanisms of heart failure. This paper is prepared carefully, the methods are described comprehensively, and the conclusions correspond to the results obtained. I have no negative comments about the work, and I believe that the manuscript can be published in its current form.
Author Response
We sincerely thank the reviewer for the favorable remarks on our study and manuscript.
Reviewer 2 Report
Comments and Suggestions for Authors
The authors conducted a study to evaluate the role of valosin-containing protein (VCP), an ATPase-associated protein, in cardiovascular pathophysiology. The authors conducted a series of experiments on mice and reported that mice with homozygous knockout (KO) of the gene were embryonically lethal. In contrast, mice with heterozygous KO, which resulted in a partial reduction of VCP in the heart, were viable at birth but progressively developed heart failure and died by 12 months of age. Suppressing VCP specifically activated mTOR complex 1 (mTORC1) early on, without affecting mTORC2. Long-term suppression of VCP led to increased expression and nuclear translocation of protein phosphatase 1 (PP1), a crucial protein dephosphorylation mediator, and was associated with reduced phosphorylation of AKTSer473 in VCP KO mouse hearts later in life, but not at an early stage. These changes over time were linked to the progressive decline in cardiac function. Suggestions:
1) Abstract: Please state your central hypothesis at the beginning of the abstract. Please show some numeric data /results in the abstract.
2) Please format the manuscript based on a standard reporting guideline, example, the ARRIVE 2.0 guideline. Please add a completed ARRIVE checklist as an appendix. https://arriveguidelines.org/resources/author-checklists
3) The methods/experiments are hard to follow. Please summarize all experiments/methods in a single figure with enough details to understand what was done (study design, samples collected, 'n' for mice/samples, etc.) which should be placed at the end of the intro or beginning of the results section.
4) Please dedicate a paragraph discussing the limitations of your study (for all experiments conducted); this para can be place just before your last/ conclusions para in the discussion section.
5) A figure summarizing the biological pathways to support your hypotheses will help improve the accessibility of the discussion section and for interpretation of your findings.
Comments on the Quality of English Languagen/a
Reviewer 3 Report
Comments and Suggestions for Authors
Comments:
This study offers a thorough investigation into the role of Valosin-containing protein (VCP) in cardiac development and function, with a particular focus on the mTOR signaling pathway. By utilizing a cardiac-specific VCP knockout (VCP KO) mouse model, the authors have shed light on how VCP deficiency affects cardiac physiology and cellular signaling under both normal and pathological conditions. The findings have significant implications for understanding the mechanisms behind cardiomyopathy and heart failure.
Strengths
-
Novelty and Importance:
- This is the first study to systematically examine the effects of VCP deficiency in the heart under normal conditions, addressing a crucial gap in the literature.
- The research underscores the essential role of VCP in maintaining cardiac function and development, highlighting its importance in cardiac biology.
-
Methodological Rigor:
- The use of a cardiac-specific VCP KO mouse model is well-chosen and justified, allowing for targeted exploration of VCP's role in the heart.
- The study employs robust methodologies, including echocardiography, histology, western blotting, and subcellular fractionation, to provide a comprehensive assessment of cardiac function and molecular changes.
-
Detailed Analysis:
- The temporal analysis of VCP’s impact on mTORC1 and mTORC2 signaling offers valuable insights into the compensatory mechanisms and long-term effects of VCP deficiency.
- Examining the subcellular distribution of VCP and PP1 at different ages adds depth to our understanding of how VCP regulates cellular processes.
Critical Comments
-
Subcellular Distribution and Functional Impact:
- While the study provides detailed analysis of the subcellular distribution of VCP and PP1, it would benefit from functional assays that directly correlate these distributions with specific cellular outcomes. For instance, assessing autophagic flux or mitochondrial function could offer more mechanistic insights into how VCP deficiency impacts cardiomyocyte health.
-
Selective Activation of mTORC1:
- The study convincingly shows that VCP deficiency leads to selective activation of mTORC1. However, further investigation into the upstream regulators of mTORC1 in the context of VCP deficiency could enhance the understanding of this selective activation. Exploring the involvement of AMPK or the TSC1/2 complex could provide additional insights.
-
Role of p47:
- The increase in p47 in VCP KO hearts is highlighted as a critical factor in the activation of PP1. While this finding is novel and interesting, functional studies demonstrating the direct interaction and impact of p47 on PP1 activity and AKT dephosphorylation in cardiomyocytes would strengthen this conclusion.
-
Longitudinal Functional Studies:
- While the study assesses cardiac function at discrete time points, continuous monitoring of cardiac function using telemetry could provide more nuanced insights into the progression of cardiac dysfunction in VCP KO mice.
-
Potential Therapeutic Implications:
- The discussion could be enriched by exploring potential therapeutic strategies targeting the pathways dysregulated by VCP deficiency. For example, considering the use of mTORC1 inhibitors or AKT activators in VCP-deficient models could add a translational aspect to the findings.
-
Control Groups:
- The use of multiple control groups (VCP f/+, MHC Cre+/-, and WT) is commendable. However, a more detailed discussion on why each control group was included and how they help isolate the effects of VCP deficiency would be beneficial.
Conclusion
This study significantly advances our understanding of the role of VCP in regulating mTOR signaling and maintaining cardiac function. The findings are robust and well-supported by comprehensive experimental data. Addressing the suggested points could further strengthen the study and provide deeper mechanistic insights, enhancing its impact and relevance to the field. Overall, this manuscript represents a valuable contribution to our knowledge of the molecular mechanisms underlying cardiomyopathy and heart failure associated with VCP deficiency.
